# Robust Implicit Networks
# via Non-Euclidean Contractions

**Saber Jafarpour**[1,*]**, Alexander Davydov**[1,*]**, Anton V. Proskurnikov**[2,3]**, and Francesco Bullo**[1]

[1] Center for Control, Dynamical Systems and Computation, University of California,
Santa Barbara, 93106-5070, USA, {saber, davydov, bullo}@ucsb.edu.
[2] Department of Electronics and Telecommunications, Politecnico di Torino, Turin, Italy;
[3] Institute for Problems in Mechanical Engineering, Russian Academy of Sciences,
St. Petersburg, Russia, anton.p.1982@ieee.org

## Abstract

Implicit neural networks, a.k.a., deep equilibrium networks, are a class of implicit-depth learning models where function evaluation is performed by solving a fixed point equation. They generalize classic feedforward models and are equivalent to infinite-depth weight-tied feedforward networks. While implicit models show improved accuracy and significant reduction in memory consumption, they can suffer from ill-posedness and convergence instability.

This paper provides a new framework, which we call Non-Euclidean Monotone Operator Network (NEMON), to design well-posed and robust implicit neural networks based upon contraction theory for the non-Euclidean norm $\ell_\infty$. Our framework includes (i) a novel condition for well-posedness based on one-sided Lipschitz constants, (ii) an average iteration for computing fixed-points, and (iii) explicit estimates on input-output Lipschitz constants. Additionally, we design a training problem with the well-posedness condition and the average iteration as constraints and, to achieve robust models, with the input-output Lipschitz constant as a regularizer. Our $\ell_\infty$ well-posedness condition leads to a larger polytopic training search space than existing conditions and our average iteration enjoys accelerated convergence. Finally, we evaluate our framework in image classification through the MNIST and the CIFAR-10 datasets. Our numerical results demonstrate improved accuracy and robustness of the implicit models with smaller input-output Lipschitz bounds. Code is available at https://github.com/davydovalexander/Non-Euclidean_Mon_Op_Net.

## 1 Introduction

Implicit neural networks are infinite-depth learning models with layers defined implicitly through a fixed-point equation. Examples of implicit neural networks include deep equilibrium models [Bai et al., 2019] and implicit deep learning models [El Ghaoui et al., 2021]. Implicit networks can be considered as generalizations of feedforward neural networks with input-injected weight tying, i.e., training parameters are transferable between layers. Indeed, in implicit networks, function evaluation is executed by solving a fixed-point equation and backpropagation is implemented by computing gradients using implicit differentiation. Due to these unique features, implicit models enjoy more flexibility and improved memory efficiency compared to traditional neural networks. At the same time, implicit networks can suffer from instability in their training due to the nonlinear nature of their fixed-point equations and can show brittle input-output behaviors due to their model flexibility.

---

[*]These authors contributed equally.

35th Conference on Neural Information Processing Systems (NeurIPS 2021).

It is known that implicit neural networks require careful tuning and initialization to avoid ill-posed training procedures. Indeed, without additional assumptions, their fixed-point equation may not have a unique solution and the numerical algorithms for finding their solutions might not converge. Several recent works in the literature have focused on studying well-posedness and convergence of the fixed-point equations of implicit networks using frameworks such as monotone operator theory [Winston and Kolter, 2020], contraction theory [El Ghaoui et al., 2021], and a mixture of both [Revay et al., 2020]. Despite several insightful results, important questions about conditions for well-posedness of implicit networks and efficient algorithms that converge to their solutions are still open.

One of the key features of implicit neural networks is their flexibility, which might come at the cost of low input-output robustness. As first noted in [Szegedy et al., 2014], the input-output behavior of deep neural networks can be vulnerable to perturbations; close enough input data can lead to completely different outputs. This lack of robustness can lead to unreliable performance of neural networks in safety-critical applications. Among several notions of robustness, the Lipschitz constant of a neural network is a coarse but rigorous measure which can be used to estimate input-output sensitivity of the network [Szegedy et al., 2014]. For this reason, there has been a growing interest in estimating the input-output Lipschitz constant of deep neural networks with respect to the $\ell_2$-norm [Fazlyab et al., 2019, Combettes and Pesquet, 2020]. However, it turns out that in some applications, the input-output Lipschitz constants with respect to non-Euclidean norms are more informative measures for studying robustness. One such application appears in the robustness analysis of neural networks with large-scale inputs under widely-distributed adversarial perturbations (examples of these adversarial perturbations can be found in [Szegedy et al., 2014]). For these examples, the input-output $\ell_2$-Lipschitz constant does not provide complete information about robustness of the network; a neural network with small input-output $\ell_2$-Lipschitz constant can be very sensitive to widespread entrywise-small perturbations of the input signal. On the other hand, the input-output $\ell_\infty$-Lipschitz constant provides a different metric which appears to be well-suited for the analysis of widespread distributed perturbations. Another application is the estimation of input signal confidence intervals from output deviations, where the input-output $\ell_\infty$-Lipschitz constant of the network provides more scalable bounds than its $\ell_2$ counterpart.

**Related works**

**Implicit learning models.**   Numerous works in learning theory have shown the power of deep learning models with implicit layers. In these learning models, the notion of layers are replaced by a composition rule, which can be either a fixed-point iteration or a solution to a differential equation. Well-known frameworks for deep learning using implicit infinite-depth layers include deep equilibrium networks [Bai et al., 2019], implicit deep learning [El Ghaoui et al., 2021], and Neural ODEs [Chen et al., 2018]. In [Kag et al., 2020], a class of implicit recurrent neural networks is considered and it is demonstrated that, with this architecture, the models do not suffer from vanishing nor exploding gradients. Implicit layers have also been used to study convex optimization problems [Agrawal et al., 2019] and to design control strategies [Amos et al., 2018]. Convergence to global minima of certain classes of implicit networks is studied in [Kawaguchi, 2021].

**Well-posedness and numerical algorithms for fixed-point equations.**   There has been a recent interest in studying well-posedness and numerical stability of implicit-depth learning models. [El Ghaoui et al., 2021] proposes a sufficient spectral condition for well-posedness and for convergence of the Picard iterations associated with the fixed-point equation of implicit networks. In [Winston and Kolter, 2020, Revay et al., 2020], using monotone operator theory, a suitable parametrization of the weight matrix is proposed which guarantees the stable convergence of suitable fixed-point iterations. A recent influential survey on monotone operators is [Ryu and Boyd, 2016]. A recent survey on fixed point strategies in data science is given by [Combettes and Pesquet, 2021].

**Robustness of learning models**   It is known that neural networks can be vulnerable to adversarial input perturbations [Szegedy et al., 2014]. A large body of literature is devoted to improve robustness of neural networks using various defense strategies against adversarial examples [Goodfellow et al., 2015, Papernot et al., 2016]. While these strategies are effective in many scenarios, they do not provide formal guarantees for robustness [Carlini and Wagner, 2017]. However, there has been a recent interest in designing classifiers that are provably robust with respect to adversarial perturbations [Madry et al., 2018, Wong and Kolter, 2018]. The input-output Lipschitz constant of a neural

network is a rigorous metric for its worst-case sensitivity with respect to input perturbations. Several recent works have focused on estimating the Lipschitz constant and enforcing its boundedness. For example, [Fazlyab et al., 2019, 2020] propose a convex optimization framework using quadratic constraints and semidefinite programming to obtain upper bounds on Lipschitz constants of deep neural networks. In [Pauli et al., 2021], a training algorithm is designed to ensure boundedness of the Lipschitz constant of the neural network via a semidefinite program. Other methods for estimating the Lipschitz constant of deep neural networks include [Krishnan et al., 2020, Revay et al., 2021, Combettes and Pesquet, 2020]. For implicit neural networks, a sensitivity-based robustness analysis is proposed in [El Ghaoui et al., 2021]. Lipschitz constants of deep equilibrium networks have also been studied in [Pabbaraju et al., 2021, Revay et al., 2020] using monotone operator theory.

#### Contributions

In this paper, using non-Euclidean contraction theory with respect to the $\ell_\infty$-norm, we propose our novel framework, Non-Euclidean Monotone Operator Network (NEMON), to design implicit neural networks and study their well-posedness, stability, and robustness. First, we develop elements of a novel non-Euclidean monotone operator theory akin to the frameworks in [Bauschke and Combettes, 2017, Ryu and Boyd, 2016]. Using the concept of matrix measure, we introduce the essential notion of one-sided Lipschitz constant of a map. Based upon this notion, we prove a general fixed-point theorem with weaker requirements than classical results on Picard and Krasnosel'skii–Mann iterations. For maps with one-sided Lipschitz constant less than unity, we show that an average iteration converges for sufficiently small step sizes and optimize its rate of convergence. For the special case of the weighted $\ell_\infty$-norm, we show that this average iteration can be accelerated by choosing a larger step size. Additionally, we study perturbed fixed-point equations and establish a bound on the distance between perturbed and nominal equilibrium points as a function of one-sided Lipschitz condition. Second, for implicit neural networks, we use our new fixed-point theorem to (i) establish $\ell_\infty$-norm conditions for their well-posedness, (ii) design accelerated numerical algorithms for computing their solutions, and (iii) provide upper bounds on their input-output $\ell_\infty$-Lipschitz constants. Third, we propose a parametrization for matrices with appropriate bound on their one-sided Lipschitz constants and use this parametrization with the average iteration to design a training optimization problem. Finally, we perform several numerical experiments illustrating improved performance of NEMON in image classification on the MNIST and the CIFAR-10 datasets compared to the state-of-the-art models in [El Ghaoui et al., 2021, Winston and Kolter, 2020]. Additionally, by adding the input-output Lipschitz constant as regularizer in the training problem, we observe improved robustness to some classes of adversarial perturbations. We include all relevant proofs in Appendix C.

## 2   Review material

**Matrix measures**   Let $\| \cdot \|$ be a norm on $\mathbb{R}^n$ and its induced norm on $\mathbb{R}^{n \times n}$. The matrix measure of $A \in \mathbb{R}^{n \times n}$ with respect to $\| \cdot \|$ is defined by $\mu(A) := \lim_{h \to 0^+} \frac{\|I_n + hA\| - 1}{h}$, that is, the one-sided directional derivative of the induced norm in direction of $A$, evaluated at $I_n$. Remarkably, the matrix measure is a tighter upper bound on the spectral abscissa of $A$ than $\|A\|$ and the set of matrices $A \in \mathbb{R}^{n \times n}$ satisfying $\mu(A) \leq 1$ is an unbounded subset of $\mathbb{R}^{n \times n}$ strictly containing the compact ball $\|A\| \leq 1$. We refer to [Desoer and Haneda, 1972] for a list of properties enjoyed by matrix measures.

We will be specifically interested in diagonally weighted $\ell_\infty$ norms defined by

$$\|x\|_{\infty,[\eta]^{-1}} = \max_i \frac{1}{\eta_i} |x_i|, \tag{1}$$

where, given a positive vector $\eta \in \mathbb{R}^n_{>0}$, we use $[\eta]$ to denote the diagonal matrix with diagonal entries $\eta$. The corresponding matrix norm and measure are

$$\|A\|_{\infty,[\eta]^{-1}} = \max_{i \in \{1,\dots,n\}} \sum_{j=1}^n \frac{\eta_j}{\eta_i} |a_{ij}|, \quad \mu_{\infty,[\eta]^{-1}}(A) = \max_{i \in \{1,\dots,n\}} \left( a_{ii} + \sum_{j=1,j \neq i}^n |a_{ij}| \frac{\eta_j}{\eta_i} \right). \tag{2}$$

**Lipschitz maps**   Given a norm $\| \cdot \|$ with induced matrix measure $\mu(\cdot)$, a differentiable map $\mathsf{F} : \mathbb{R}^n \to \mathbb{R}^n$ is Lipschitz continuous with constant $\mathrm{Lip}(\mathsf{F}) \in \mathbb{R}_{\geq 0}$ if

$$\|D\mathsf{F}(x)\| \leq \mathrm{Lip}(\mathsf{F}) \qquad \text{for all } x \in \mathbb{R}^n. \tag{3}$$

For example, for an affine $\mathsf{F}(x) = Ax + b$, the (smallest) Lipschitz constant is $\mathrm{Lip}(\mathsf{F}) = \|A\|$.

**One-sided Lipschitz maps**   Given a norm $\| \cdot \|$, a differentiable map $\mathsf{F} : \mathbb{R}^n \to \mathbb{R}^n$ is one-sided Lipschitz continuous with constant $\mathrm{osL}(\mathsf{F}) \in \mathbb{R}$ if

$$\mu(D\mathsf{F}(x)) \leq \mathrm{osL}(\mathsf{F}) \qquad \text{for all } x \in \mathbb{R}^n. \tag{4}$$

For example, for an affine $\mathsf{F}(x) = Ax + b$, the (smallest) one-sided Lipschitz constant is $\mathrm{osL}(\mathsf{F}) = \mu(A)$. Note that (i) the one-sided Lipschitz constant is upper bounded by the Lipschitz constant, (ii) a Lipschitz continuous map is always one-sided Lipschitz continuous, and (iii) the one-sided Lipschitz constant may be negative. For a more in-depth review we refer to Appendix A. The notion of one-sided Lipschitz continuity unifies several important concepts in dynamical systems and optimization theory. In operator theory, the map $\mathsf{F}$ is called a monotone operator if it is one-sided Lipschitz continuous with respect to the $\ell_2$-norm with the constant $-\mathrm{osL}(-\mathsf{F}) > 0$ [Ryu and Boyd, 2016, Bauschke and Combettes, 2017]. In control theory, the vector field $\mathsf{F}$ is called strongly infinitesimally contracting if it is one-sided Lipschitz continuous with the constant $\mathrm{osL}(\mathsf{F}) < 0$ [Desoer and Haneda, 1972, Lohmiller and Slotine, 1998, Pavlov et al., 2004]. In what follows, we let $\mathrm{osL}_{\infty,[\eta]^{-1}}(\mathsf{F}) \in \mathbb{R}$ denote the one-sided Lipschitz constants with respect to the weighted $\ell_\infty$-norm.

## 3   Fixed-point equations and one-sided Lipschitz constants

In this section, we show that the notion of one-sided Lipschitz constant can be used to study solvability of fixed-point equation:

$$x = \mathsf{F}(x), \tag{5}$$

where $\mathsf{F} : \mathbb{R}^n \to \mathbb{R}^n$ is a differentiable map. Let $\| \cdot \|$ be a norm on $\mathbb{R}^n$, then in view of the Banach fixed-point theorem, a simple sufficient condition for existence of a unique solution for the fixed-point equation (5) is $\mathrm{Lip}(\mathsf{F}) < 1$. We note that the sufficient condition $\mathrm{Lip}(\mathsf{F}) < 1$ depends on the specific form of the fixed-point equation (5) and can be relaxed by a suitable rewriting of this fixed-point equation. Given an averaging parameter $\alpha \in (0, 1]$ we define the average map $\mathsf{F}_\alpha : \mathbb{R}^n \to \mathbb{R}^n$ by $\mathsf{F}_\alpha := (1 - \alpha)\mathsf{I} + \alpha\mathsf{F}$, where $\mathsf{I}$ is the identity map. Using this notion, an equivalent reformulation of the fixed-point equation (5) is:

$$x = (1 - \alpha)x + \alpha\mathsf{F}(x) = \mathsf{F}_\alpha(x). \tag{6}$$

For $\alpha = 1$, we have $\mathsf{F}_\alpha(x) = \mathsf{F}(x)$ and equation (6) coincides with equation (5). For every $\alpha \in (0, 1)$, the map $\mathsf{F}_\alpha$ is different from $\mathsf{F}$ but equations (5) and (6) are equivalent. Hence, if $\mathrm{Lip}(\mathsf{F}_\alpha) < 1$, then by the Banach fixed-point theorem, the fixed point equation (6) (and therefore the fixed point equation (5)) has a unique solution $x^*$ and the sequence $\{y_k\}_{k=1}^\infty$ defined by

$$y_{k+1} = (1 - \alpha)y_k + \alpha\mathsf{F}(y_k), \qquad \text{for all } k \in \mathbb{Z}_{\geq 0} \tag{7}$$

converges geometrically to $x^*$ with rate $\mathrm{Lip}(\mathsf{F}_\alpha)$. As a result of the parametrization (6), the condition $\mathrm{Lip}(\mathsf{F}) < 1$ for existence and uniqueness of the fixed-point can be relaxed to sufficient conditions

$$\mathrm{Lip}(\mathsf{F}_\alpha) < 1, \tag{8}$$

parametrized by $\alpha \in (0, 1]$. Additionally, if condition (8) is satisfied, then algorithm (7) computes the fixed point $x^*$. It can be shown that the condition (8) becomes less conservative as $\alpha$ decreases. The next theorem shows that in the limit as $\alpha \to 0^+$, condition (8) approaches the condition $\mathrm{osL}(\mathsf{F}) < 1$.

**Theorem 1** (Fixed points via one-sided Lipschitz conditions)**.** *Let $\mathsf{F} : \mathbb{R}^n \to \mathbb{R}^n$ be differentiable and Lipschitz with constant $\ell > 0$ with respect to a norm $\| \cdot \|$. Define the average map $\mathsf{F}_\alpha = (1 - \alpha)\mathsf{I} + \alpha\mathsf{F}$ and, for $c > 0$, the function $\gamma_{\ell,c} : \,]0, \frac{c}{(c+\ell+1)(\ell+1)}[ \,\to \mathbb{R}$ by:*

$$\gamma_{\ell,c}(\alpha) := \left(1 + \alpha c - \frac{\alpha^2(\ell+1)^2}{1 - \alpha(\ell+1)}\right)^{-1}.$$

*Then the following statements are equivalent:*

*(i)* $\mathrm{osL}(\mathsf{F}) < 1 - c$,

*(ii)* $\mathrm{Lip}(\mathsf{F}_\alpha) = \gamma_{\ell,c}(\alpha)$, *for* $0 < \alpha < \frac{c}{(c+\ell+1)(\ell+1)}$.

*Moreover, if the equivalent conditions (i) or (ii) hold, then, for condition number $\kappa = \frac{1+\mathrm{Lip}(\mathsf{F})}{1-\mathrm{osL}(\mathsf{F})}$,*

*(iii)* $\mathsf{F}$ *has a unique fixed point $x^*$;*

*(iv) for $0 < \alpha < \frac{1}{\kappa(\kappa+1)}$, $\mathsf{F}_\alpha$ is a contraction mapping with contraction factor $\gamma_{\ell,c}(\alpha) < 1$;*

*(v) the minimum contraction factor $\gamma_{\ell,c}^* = 1 - \frac{1}{4\kappa^2} + \frac{1}{8\kappa^3} + \mathcal{O}\left(\frac{1}{\kappa^4}\right)$ and the minimizing averaging parameter $\alpha^*$ of $\mathsf{F}_\alpha$ is*

$$\alpha^* = \frac{\kappa}{1 - \mathrm{osL}(\mathsf{F})}\left(1 - \frac{1}{\sqrt{1 + 1/\kappa}}\right) = \frac{1}{1 - \mathrm{osL}(\mathsf{F})}\left(\frac{1}{2\kappa^2} - \frac{3}{8\kappa^3} + \mathcal{O}\left(\frac{1}{\kappa^4}\right)\right).$$

The average iteration (6) is often referred to as the Krasnosel'skii–Mann iteration or the damped iteration [Bauschke and Combettes, 2017]. Compared to [Bauschke and Combettes, 2017, Theorem 5.15], Theorem 1(iv) studies convergence of the Krasnosel'skii–Mann iteration for arbitrary norms, proposes a weaker convergence condition of the form $\mathrm{osL}(\mathsf{F}) < 1$ (hence, $\mathsf{F}$ need not be non-expansive). However, it ensures convergence for only sufficiently small $\alpha > 0$ and assumes that $\mathsf{F}$ is differentiable (as will be shown, however, the latter assumption can be relaxed).

## 3.1 Accelerated convergence for weighted $\ell_\infty$ norms

For diagonally weighted $\ell_\infty$ norms, one can strengthen Theorem 1(iv) to prove the convergence of the average iteration (6) on a larger domain of the parameter $\alpha$.

**Theorem 2** (Accelerated fixed point algorithm for $\ell_\infty$ norms). *Let $\mathsf{F} : \mathbb{R}^n \to \mathbb{R}^n$ be differentiable and Lipschitz with respect to the weighted non-Euclidean norm $\|\cdot\|_{\infty,[\eta]^{-1}}$. Define the average map $\mathsf{F}_\alpha = (1-\alpha)\mathsf{I} + \alpha\mathsf{F}$ and pick $\mathrm{diagL}(\mathsf{F}) \in [-\mathrm{Lip}(\mathsf{F}), \mathrm{osL}(\mathsf{F})]$ to satisfy*

$$\mathrm{diagL}(\mathsf{F}) \le \min_{i \in \{1,\dots,n\}} \inf_{x \in \mathbb{R}^n} D\mathsf{F}_{ii}(x). \tag{9}$$

*If $\mathrm{osL}(\mathsf{F}) < 1$, then $\mathsf{F}$ has a unique fixed-point $x^*$ and*

*(i) for $0 < \alpha \le \dfrac{1}{1 - \mathrm{diagL}(\mathsf{F})}$, $\mathsf{F}_\alpha$ is a contraction mapping with the contraction factor $1 - \alpha(1 - \mathrm{osL}(\mathsf{F})) < 1$;*

*(ii) the minimum contraction factor and minimizing averaging parameter of $\mathsf{F}_\alpha$ are, respectively,*

$$\mathrm{Lip}(\mathsf{F}_{\alpha^*}) = 1 - \frac{1 - \mathrm{osL}(\mathsf{F})}{1 - \mathrm{diagL}(\mathsf{F})} = 1 - \frac{1}{\kappa_\infty}, \qquad for\ \kappa_\infty = \frac{1 - \mathrm{diagL}(\mathsf{F})}{1 - \mathrm{osL}(\mathsf{F})} \le \frac{1 + \mathrm{Lip}(\mathsf{F})}{1 - \mathrm{osL}(\mathsf{F})},$$
$$\alpha^* = \frac{1}{1 - \mathrm{diagL}(\mathsf{F})}.$$

Note that $\mathrm{diagL}(\mathsf{F})$ is well-defined because of the Lipschitz continuity assumption. Specifically, one can show that $\mathrm{diagL}(\mathsf{F})$ is the minus the minimum over $i \in \{1, \dots, n\}$ of the one-sided Lipschitz constants of the maps $x_i \mapsto -\mathsf{F}(x_i, x_{-i})$ at $x_{-i} = (x_1, \dots, x_{i-1}, x_{i+1}, \dots, x_n)$ fixed.

It is instructive to compare the minimum contraction factor in the general Theorem 1 with the minimum contraction factor for $\ell_\infty$ norms in Theorem 2 and how they depend upon the corresponding condition numbers $\kappa$ and $\kappa_\infty$. We note that (i) the relevant condition number diminishes $\kappa \ge \kappa_\infty$, and (ii) the minimum contraction factor $\mathrm{Lip}(\mathsf{F}_{\alpha^*}) = 1 - \frac{1}{4\kappa^2} + \mathcal{O}(1/\kappa^4)$ improves to $\mathrm{Lip}(\mathsf{F}_{\alpha^*}) = 1 - \frac{1}{\kappa_\infty}$. This acceleration justifies the title of this section.

## 3.2 Perturbed fixed-point problems

In this subsection, we focus on solvability of the perturbed fixed-point equation:

$$x = \mathsf{F}(x, u), \tag{10}$$

where $\mathsf{F} : \mathbb{R}^n \times \mathbb{R}^r \to \mathbb{R}^n$ is differentiable in $x$. We define $\mathsf{F}_u(x) = \mathsf{F}(x, u)$ and $\mathsf{F}_x(u) = \mathsf{F}(x, u)$. Given a norm $\|\cdot\|_{\mathcal{X}}$ in $\mathbb{R}^n$ and $\|\cdot\|_{\mathcal{U}}$ in $\mathbb{R}^r$, $\mathsf{F}$ is Lipschitz in its first argument with constant $\mathrm{Lip}_x(\mathsf{F}) \in \mathbb{R}_{\ge 0}$ if

$$\|\mathsf{F}(x_1, u) - \mathsf{F}(x_2, u)\|_{\mathcal{X}} \le \mathrm{Lip}_x(\mathsf{F})\|x_1 - x_2\|_{\mathcal{X}} \quad \text{for all } x_1, x_2 \in \mathbb{R}^n \text{ and } u \in \mathbb{R}^r,$$

and it is Lipschitz in its second argument with constant $\mathrm{Lip}_u(\mathsf{F}) \in \mathbb{R}_{\geq 0}$ if

$$\|\mathsf{F}(x, u_1) - \mathsf{F}(x, u_2)\|_{\mathcal{X}} \leq \mathrm{Lip}_u(\mathsf{F})\|u_1 - u_2\|_{\mathcal{U}} \quad \text{for all } x \in \mathbb{R}^n \text{ and } u_1, u_2 \in \mathbb{R}^r,$$

and it is one-sided Lipschitz in its first argument with constant $\mathrm{osL}_x(\mathsf{F}) \in \mathbb{R}$ if

$$\mu(D_x\mathsf{F}(x, u)) \leq \mathrm{osL}_x(\mathsf{F}) \quad \text{for all } x_1, x_2 \in \mathbb{R}^n \text{ and } u \in \mathbb{R}^r.$$

The following result, which is in the spirit of Lim's Lemma [Lim, 1985], provides an upper bound on the distance between fixed-points of the perturbed equation (10).

**Theorem 3** (Perturbed fixed-points). *Given a norm $\|\cdot\|_{\mathcal{X}}$ in $\mathbb{R}^n$ and a norm $\|\cdot\|_{\mathcal{U}}$ in $\mathbb{R}^r$, consider a map $\mathsf{F} : \mathbb{R}^n \times \mathbb{R}^r \to \mathbb{R}^n$ differentiable in the first argument and Lipschitz in both arguments. If $\mathsf{F}$ is one-sided Lipschitz with constant $\mathrm{osL}_x(\mathsf{F}) < 1$, then*

*(i) for every $u \in \mathbb{R}^m$, the map $\mathsf{F}_u$ has a unique fixed point $x_u^*$;*

*(ii) for every $u, v \in \mathbb{R}^m$, $\|x_u^* - x_v^*\|_{\mathcal{X}} \leq \dfrac{\mathrm{Lip}_u(\mathsf{F})}{1 - \mathrm{osL}_x(\mathsf{F})}\|u - v\|_{\mathcal{U}}.$*

Finally, Theorems 1, 2, and 3 are not directly applicable to activation function that are not differentiable. In Appendix C.3, we show that for specific form of the fixed-point equation (5), where $\mathsf{F} = \Phi \circ \mathsf{H}$ and $\Phi : \mathbb{R}^n \to \mathbb{R}^n$ is a weakly increasing, non-expansive, diagonal activation function and $\mathsf{H} : \mathbb{R}^n \times \mathbb{R}^r \to \mathbb{R}^n$ is a differentiable function, all of the conclusions of Theorems 1, 2, and 3 hold by requiring equation (9) to be true almost everywhere.

# 4 Contraction analysis of implicit neural networks

In this section, we use contraction theory to lay the foundation for our Non-Euclidean Monotone Operator Network (NEMON) model of implicit neural networks. Given $A \in \mathbb{R}^{n \times n}$, $B \in \mathbb{R}^{n \times r}$, $C \in \mathbb{R}^{q \times n}$, and $D \in \mathbb{R}^{q \times r}$, we consider the implicit neural network

$$x = \Phi(Ax + Bu) := \mathsf{N}(x, u), \qquad y = Cx + Du, \tag{11}$$

where $x \in \mathbb{R}^n$, $u \in \mathbb{R}^r$, $y \in \mathbb{R}^q$, and $\Phi : \mathbb{R}^n \to \mathbb{R}^n$ is defined by $\Phi(x) = (\phi_1(x_1), \ldots, \phi_n(x_n))$. For every $i \in \{1, \ldots, n\}$, we assume the activation function $\phi_i : \mathbb{R} \to \mathbb{R}$ is weakly increasing, i.e., $\phi_i(x_i) \geq \phi_i(z_i)$ for $x_i \geq z_i$, and non-expansive, i.e., $|\phi_i(x_i) - \phi_i(z_i)| \leq |x_i - z_i|$ for all $x_i$ and $z_i$; if $\phi_i$ is differentiable, these conditions are equivalent to $0 \leq \phi_i'(x_i) \leq 1$ for all $x_i \in \mathbb{R}$.

We are able to provide the following estimates on all relevant Lipschitz constants.

**Theorem 4** (Lipschitz and one-sided Lipschitz constants for the implicit neural network). *Consider the implicit neural network in equation (11) with weakly increasing and non-expansive activation functions $\Phi$. With respect to $\|\cdot\|_{\infty, [\eta]^{-1}}$, $\eta \in \mathbb{R}_{>0}^n$, on $\mathbb{R}^n$ and $\|\cdot\|_{\mathcal{U}}$ on the input space $\mathbb{R}^r$, the map $\mathsf{N} : \mathbb{R}^n \times \mathbb{R}^r \to \mathbb{R}^n$ is one-sided Lipschitz continuous in the first variable and Lipschitz continuous in both variables with constants:*

$$\mathrm{osL}_x(\mathsf{N}) = \mu_{\infty, [\eta]^{-1}}(A)_+ , \qquad \mathrm{Lip}_x(\mathsf{N}) = \|A\|_{\infty, [\eta]^{-1}} , \tag{12}$$

$$\mathrm{Lip}_u(\mathsf{N}) = \|B\|_{(\infty, [\eta]^{-1}), \mathcal{U}} , \qquad \mathrm{diagL}(\mathsf{N}) = \min_{i \in \{1, \ldots, n\}}(A_{ii})_- , \tag{13}$$

*where $(z)_+ = z$ if $z \geq 0$ and $(z)_+ = 0$ if $z < 0$; and $(z)_- = 0$ if $z \geq 0$ and $(z)_- = z$ if $z < 0$.*

We now use these estimates to establish multiple properties of the implicit neural network.

**Corollary 5** (Well posedness, input-state Lipschitz constant, and computation). *Consider the model (11), with parameters $(A, B, C, D)$ and with weakly increasing and non-expansive activation functions $\Phi$. Define the average map $\mathsf{N}_\alpha := (1 - \alpha)\mathsf{I} + \alpha\mathsf{N}$ and consider the norms $\|\cdot\|_{\infty, [\eta]^{-1}}$, $\eta \in \mathbb{R}_{>0}^n$, on $\mathbb{R}^n$, $\|\cdot\|_{\mathcal{U}}$ on the input space $\mathbb{R}^r$ and $\|\cdot\|_{\mathcal{Y}}$ on the output space $\mathbb{R}^q$. Then*

*(i) if $\mu_{\infty, [\eta]^{-1}}(A) < 1$, then (11) is well posed, i.e., there exists a unique fixed point,*

*(ii) the map $\mathsf{N}_\alpha$ is a contraction mapping for $0 < \alpha \leq \alpha^* := \left(1 - \min_{i \in \{1, \ldots, n\}}(A_{ii})_-\right)^{-1}$ with minimum contraction factor $\mathrm{Lip}(\mathsf{N}_{\alpha^*}) = 1 - \dfrac{1 - \mu_{\infty, [\eta]^{-1}}(A)_+}{1 - \min_{i \in \{1, \ldots, n\}}(A_{ii})_-}.$*

*(iii) the Lipschitz constants from input $u$ to fixed point $x_u^*$ and to the output $y = Cx_u^* + Du$ are*

$$\text{Lip}_{u \to x^*} := \frac{\text{Lip}_u(\mathsf{N})}{1 - \text{osL}_x(\mathsf{N})} = \frac{\|B\|_{(\infty,[\eta]^{-1}),\mathcal{U}}}{1 - \mu_{\infty,[\eta]^{-1}}(A)_+}, \tag{14}$$

$$\text{Lip}_{u \to y} := \frac{\|B\|_{(\infty,[\eta]^{-1}),\mathcal{U}}\|C\|_{\mathcal{Y},(\infty,[\eta]^{-1})}}{1 - \mu_{\infty,[\eta]^{-1}}(A)_+} + \|D\|_{\mathcal{Y},\mathcal{U}}. \tag{15}$$

## 5 Training implicit neural networks

**Problem setup** Given an input data matrix $U = [u_1, \ldots, u_m] \in \mathbb{R}^{r \times m}$ and a corresponding output data matrix $Y = [y_1, \ldots, y_m] \in \mathbb{R}^{q \times m}$, we aim to learn matrices $A, B, C, D$ so that the neural network (11) approximates the input-output relationship. We rewrite the model for matrix inputs as $\widehat{Y} = CX + DU$, where $X = \Phi(AX + BU)$. From Corollary 5(i), if each $\phi_i$ is weakly increasing and non-expansive, the fixed point problem is well-posed when $\mu_{\infty,[\eta]^{-1}}(A) < 1$ for some $\eta \in \mathbb{R}^n_{>0}$. We consider a training problem of the form

$$\min_{A,B,C,D,X} \quad \mathcal{L}(Y, CX + DU) + \mathcal{P}(A, B, C, D)$$
$$X = \Phi(AX + BU), \quad \mu_{\infty,[\eta]^{-1}}(A) \leq \gamma, \tag{16}$$

where $\mathcal{L}$ is a loss function, $\mathcal{P}$ is a penalty function, and $\gamma < 1$ is a hyperparameter ensuring the fixed point problem is well-posed. For $\eta = \mathbb{1}_n$, we can remove the constraint $\mu_\infty(A) \leq \gamma$ in the training optimization problem (16) using the following parametrization of weight matrix $A$:

$$A = T - \text{diag}(|T|\mathbb{1}_n) + \gamma I_n. \tag{17}$$

In Appendix B, we show that parametrization (17) characterizes the set of matrices in $\mathbb{R}^{n \times n}$ satisfying $\mu_\infty(A) \leq \gamma$. Using the parametrization (17) in the training problem not only improves the computational efficiency of the optimization but also allows for the design of implicit neural networks with additional structure such as convolutions. Suppose $u \in \mathbb{R}^{rs^2}$ is a $r$-channel input of size $s \times s$ and $x \in \mathbb{R}^{ns^2}$ is an $n$-channel hidden layer. To define our implicit CNN, we select the weight matrix $A \in \mathbb{R}^{ns^2 \times ns^2}$ as the matrix form of a 2D convolutional operator. If we consider a circular convolution operator, then $A$ is a circulant matrix. Using the parametrization (17), $A$ is circulant if and only if $T$ is circulant. Therefore, the training problem for implicit CNNs can be cast as an unconstrained optimization problem using the above parametrization with a circulant $T$.

**Improving robustness via Lipschitz regularization** We now focus on learning robust implicit neural networks with bounded Lipschitz constants via a regularization strategy. Setting both $\|\cdot\|_\mathcal{U}$ and $\|\cdot\|_\mathcal{Y}$ as $\|\cdot\|_\infty$ in the input-output Lipschitz bound (15), we get

$$\text{Lip}_{u \to y} = \frac{\|B\|_{(\infty,[\eta]^{-1}),(\infty)}\|C\|_{(\infty),(\infty,[\eta]^{-1})}}{1 - \mu_{\infty,[\eta]^{-1}}(A)_+} + \|D\|_{\infty,\infty}$$
$$\leq \frac{1}{2}\frac{\|B\|^2_{(\infty,[\eta]^{-1}),(\infty)} + \|C\|^2_{(\infty),(\infty,[\eta]^{-1})}}{1 - \mu_{\infty,[\eta]^{-1}}(A)_+} + \|D\|_{\infty,\infty},$$

where the inequality provides a convex upper bound for the input-output Lipschitz constant. Therefore, using the hyperparameter $\lambda > 0$, the regularized optimization problem is written as

$$\min_{A,B,C,D,X} \quad \mathcal{L}(Y, CX + DU) + \lambda\Big(\frac{1}{2}\frac{\|B\|^2_{(\infty,[\eta]^{-1}),(\infty)} + \|C\|^2_{(\infty),(\infty,[\eta]^{-1})}}{1 - \mu_{\infty,[\eta]^{-1}}(A)_+} + \|D\|_{\infty,\infty}\Big)$$
$$X = \Phi(AX + BU), \quad \mu_{\infty,[\eta]^{-1}}(A) \leq \gamma. \tag{18}$$

**Certified adversarial robustness via Lipschitz bounds** Given a nominal input $u \in \mathbb{R}^r$, we consider any perturbed input $v$ within an $\ell_\infty$-ball of radius $\varepsilon$ around $u$. In this case, we have

$$\|y_u - y_v\|_\infty \leq \text{Lip}_{u \to y}\|u - v\|_\infty \leq \text{Lip}_{u \to y}\varepsilon. \tag{19}$$

Then we define $\text{margin}(u) = (y_u)_i - \max_{j \neq i}(y_u)_j$, where $(y_u)_i$ is the logit corresponding to the (correct) label $i$ for the input $u$. Then provided $L\varepsilon \leq \frac{1}{2}\text{margin}(u)$, NEMON is certifiably robust to any perturbed input $v$ within an $\ell_\infty$-ball of radius $\varepsilon$ centered at $u$.

**Backpropagation of gradients via average iteration** From [El Ghaoui et al., 2021] we now show how the average iteration can be used to perform backpropagation via the implicit function theorem. For simplicity, we assume that each activation function $\phi_i$ is differentiable and consider mini-batches of size 1, i.e., we have $X = x \in \mathbb{R}^n$, $U = u \in \mathbb{R}^r$ and $\widehat{Y} = \widehat{y} \in \mathbb{R}^q$. Let $x^*$ be the unique solution of the fixed-point equation (11). Then the chain rule implies

$$\frac{\partial \mathcal{L}}{\partial A} = (\nabla_{x^*}\mathcal{L})x^\top, \qquad \frac{\partial \mathcal{L}}{\partial B} = (\nabla_{x^*}\mathcal{L})u^\top,$$

$$\frac{\partial \mathcal{L}}{\partial C} = (\nabla_{\widehat{y}}\mathcal{L})x^\top, \qquad \frac{\partial \mathcal{L}}{\partial D} = (\nabla_{\widehat{y}}\mathcal{L})u^\top.$$

Since $\mathcal{L}$ depends explicitly on $\widehat{y}$, computing $\nabla_{\widehat{y}}\mathcal{L}$ is straightforward. Computing $\nabla_{x^*}\mathcal{L}$ is more complicated since $X^*$ is defined only implicitly. However, it be shown that

$$\nabla_{x^*}\mathcal{L} = (C(I - D\Phi A)^{-1}D\Phi)^\top \nabla_{\widehat{y}}\mathcal{L}.$$

Since $\mu_{\infty,[\eta]^{-1}}(A) < 1$, by Lemma 8 we get that $\mu_{\infty,[\eta]^{-1}}(D\Phi A) < 1$. This implies that the matrix $G := (I_n - D\Phi A)^{-1}D\Phi \in \mathbb{R}^{n \times n}$ exists and is the solution to the following fixed-point equation [El Ghaoui et al., 2021, Section 6.2]

$$G = D\Phi(AG + I_n). \tag{20}$$

Moreover, $\mu_{\infty,[\eta]^{-1}}(D\Phi A) < 1$ and Theorem 2 together imply that the fixed-point equation (20) has a unique solution $G^*$ and, for every $0 < \alpha \le \alpha^* := \left(1 - \min_i(A_{ii})_-\right)^{-1}$, the average iterations

$$G_{k+1} = (1 - \alpha)G_k + \alpha D\Phi(AG_k + I_n), \qquad \text{for all } k \in \mathbb{Z}_{\ge 0}$$

are contracting with the minimum contraction factor $1 - \alpha^*\left(1 - \mu_{\infty,[\eta]^{-1}}(A)_+\right)$ at step size $\alpha^*$.

## 6 Theoretical and numerical comparisons

In this section, we provide a comprehensive comparison of our framework with the state-of-the-art implicit neural networks[2].

### 6.1 Implicit network models

We start by reviewing the existing models for implicit networks in the literature.

**Implicit deep learning model.** [El Ghaoui et al., 2021] proposes a class of implicit neural networks with input-output behavior described by (11). It is shown that a sufficient condition for existence and uniqueness of a solution and convergence of the Picard iterations for the fixed point equation $x = \Phi(Ax + Bu)$ is $\lambda_{\mathrm{pf}}(|A|) < 1$, where $|A|$ denotes the entrywise absolute value of the matrix $A$ and $\lambda_{\mathrm{pf}}$ denotes the Perron-Frobenius eigenvalue. For training, the optimization problem (16) is used where the constraint $\mu_{\infty,[\eta]^{-1}}(A) \le \gamma$ is replaced by $\|A\|_\infty \le \gamma$ [El Ghaoui et al., 2021, Equation 6.3].[3] It is easy to see that our well-posedness condition in Corollary 5(i) is less conservative than $\lambda_{\mathrm{pf}}(|A|) < 1$ and its convex relaxation $\|A\|_\infty < 1$.

**Monotone operator deep equilibrium network (MON).** [Winston and Kolter, 2020] proposes to use monotone operator theory to guarantee well-posedness of the fixed-point equation as well as its convergence to the solutions. The input-output behavior of the network is described by (11). For training, the optimization problem (16) is used where the constraint $\mu_{\infty,[\eta]^{-1}}(A) \le \gamma$ is replaced by $I_n - \frac{1}{2}(A + A^\top) \succeq (1 - \gamma)I_n$. In order to ensure that this constraint is always satisfied in the training procedure, the weight matrix $A$ is parametrized as $A = \gamma I_n - W^\top W - Z + Z^\top$, for arbitrary $W, Z \in \mathbb{R}^{n \times n}$ [Winston and Kolter, 2020, Appendix D].[4] In the context of contraction theory,

$$I_n - \tfrac{1}{2}(A + A^\top) \succeq (1 - \gamma)I_n \quad \Longleftrightarrow \quad \mu_2(A) \le \gamma,$$

which is shown in Appendix A. Thus, the parametrization $A = \gamma I_n - W^\top W - Z + Z^\top$ can be considered as the $\ell_2$-norm version of the parametrization described by equation (17). In other words, the monotone operator network formulation is a Euclidean transcription of the framework we propose in this paper.

---

[2]All models were trained using Google Colab with a Tesla P100-PCIE-16GB GPU.

[3]The implicit deep learning implementation is available at `https://github.com/beeperman/idl`.

[4]The MON implementation is available at `https://github.com/locuslab/monotone_op_net`.

## 6.2 MNIST experiments

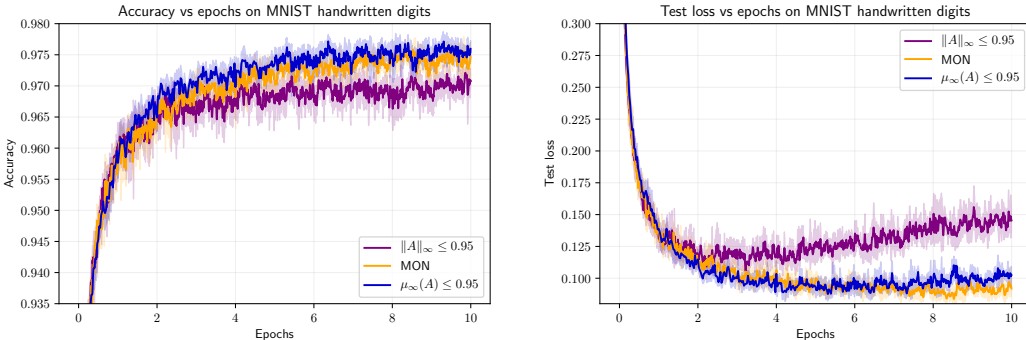

Figure 1: Performance comparison between the NEMON model with $\mu_\infty(A) \leq 0.95$, the implicit deep learning model with $\|A\|_\infty \leq 0.95$, and MON with $I_n - \frac{1}{2}(A + A^\top) \succeq 0.05 I_n$ on the MNIST dataset. The curves are generated by mean accuracy and mean loss over 5 different runs while light envelopes around the curves correspond to the standard deviation over the runs. Average best accuracy for the NEMON model is 0.9772, while it is 0.9721 for implicit deep learning model and 0.9762 for the MON model.

In the digit classification dataset MNIST, input data are $28 \times 28$ pixel images of handwritten digits between 0-9. There are 60000 training images and 10000 test images. For training, images are reshaped into 784 dimensional column vectors and entries are scaled into the range $[0, 1]$. As a loss function, we use the cross-entropy. All models are of order $n = 100$, used the ReLU activation function $\phi_i(x) = (x)_+$, and are trained with a batch size of 300 over 10 epochs with a learning rate of $1.5 \times 10^{-2}$. Curves for accuracy and loss versus epochs for the three models are shown in Figure 1. Regarding training times, using the average iteration, NEMON took, on average, 12 forward iterations, 13 backward iterations, and 9.8 seconds to train per epoch. Using the Peaceman-Rachford iteration, MON took, on average, 17 forward iterations, 16 backward iterations, and 9.5 seconds to train per epoch. Using the Picard iteration, the implicit deep learning model took, on average, 10 forward iterations, 5 backward iterations, and 5.8 seconds to train per epoch. We observe that the NEMON model performs better than the implicit deep learning model and has a comparable performance to MON.

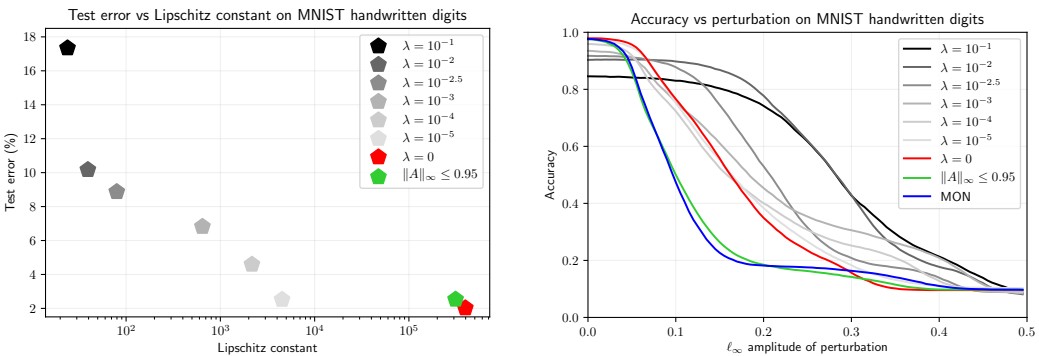

Figure 2: On the left is a plot of test error versus Lipschitz constant for the implicit deep learning model with $\|A\|_\infty \leq 0.95$ and for NEMON with $\mu_\infty(A) \leq 0.95$ and parametrized by the regularization hyperparameter $\lambda$. We define the test error as 1 minus the accuracy. On the right is a plot of accuracy versus $\ell_\infty$ perturbation of a deterministic adversarial image inversion attack where we additionally include the MON model with $I_n - \frac{1}{2}(A + A^\top) \succeq 0.05 I_n$.

We also study the robustness of the NEMON model compared to the implicit deep learning model and the MON model on the MNIST dataset. We train various models regularized by the input-output Lipschitz constant as in (18). Additionally, to verify robustness of the different models, we consider several adversarial attacks and plot the accuracy versus perturbation of such an attack. In Figure 2, we consider a continuous image inversion attack [Hosseini et al., 2017], where each pixel is perturbed

in the direction of pixel value inversion with amplitude given by the $\ell_\infty$ perturbation. For more details on this and other types of adversarial perturbations, we refer to Appendix D. We observe that for $\lambda = 10^{-5}$, the regularized NEMON model achieves a two order of magnitude decrease in its input-output Lipschitz constant compared to the un-regularized NEMON models. In addition, we see that the implicit deep learning model and the MON model are more sensitive to the continuous image inversion attack than NEMON. Moreover, as the regularization parameter $\lambda$ increases, the NEMON model becomes increasingly robust to this attack.

### 6.3 CIFAR-10 experiments

In the image classification dataset CIFAR-10, input data are $32 \times 32$ color images in 10 classes. There are 50000 training images and 10000 test images. We compare our proposed NEMON model with a convolutional structure to a single convolutional layer MON model. Each model used 81 channels. We train both models with a batch size of 256 and a learning rate of $10^{-3}$ for 40 epochs. For training, using the average iteration, NEMON took, on average, 10 forward iterations, 10 backward iterations, and 75.0 seconds per epoch to train. Using the Peaceman-Rachford iteration, MON took, on average, 5 forward iterations, 5 backward iterations, and 101.8 seconds per epoch to train.

We focus primarily on the robustness of NEMON and MON with respect to $\ell_\infty$-norm bounded perturbations on CIFAR-10. To this end, we additionally trained two NEMON models with regularization parameters $\lambda \in \{10^{-4}, 10^{-5}\}$. In Figure 3, on the left is a plot of the certified robustness of each of the models via their $\ell_\infty$-Lipschitz constants. For MON, we got the $\ell_\infty$-Lipschitz bound using the method in [Pabbaraju et al., 2021] for the $\ell_2$-Lipschitz bound and using the upper bound $\|u\|_2 \leq \sqrt{rs^2}\|u\|_\infty$. On the right is a plot of the accuracy of different models with respect to the projected gradient descent attack. We observe that the un-regularized and regularized NEMON models are more robust to $\ell_\infty$-norm bounded perturbations than is MON.

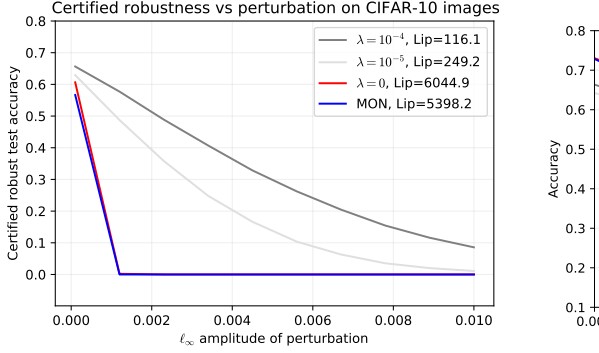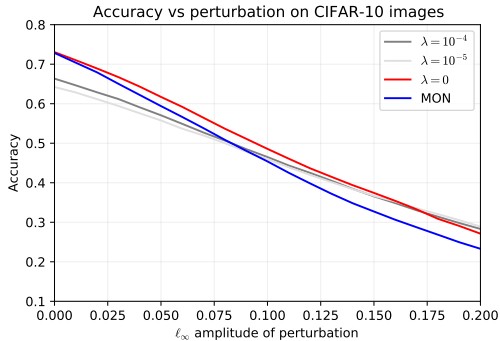

Figure 3: On the left is a plot of certified robustness via the Lipschitz constants of MON with the constraint $I_n - \frac{1}{2}(A + A^\top) \succeq I_n$ and NEMON with the constraint $\mu_\infty(A) \leq 0$. On the right is a plot of accuracy versus $\ell_\infty$ perturbation of the projected gradient descent attack.

## 7 Conclusion

Using non-Euclidean contraction theory, we propose a framework to study stability of fixed-point equations. We apply this framework to analyze well-posedness and convergence of implicit neural networks and to design an efficient training algorithm to incorporate robustness guarantees. For future research, we envision that our framework is applicable to study stability and robustness of implicit learning models with additional structure such as graph neural networks.

## 8 Acknowledgments

The authors thank Ian Manchester for stimulating discussions about contraction theory and the anonymous reviewers for their insightful feedback. This work was supported in part by DTRA under grant HDTRA1-19-1-0017, AFOSR under grant FA9550-22-1-0059, and NSF Graduate Research Fellowship under grant 2139319.

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
