# OpenReview forum: "Robust Implicit Networks via Non-Euclidean Contractions"
_NeurIPS.cc/2021/Conference — NeurIPS 2021 Poster_

### Official Review · Reviewer_ivrZ · 2021-07-13

**Rating:** 7
**Confidence:** 4

**Summary:**

This paper studies the implicit nn with non-Euclidean norms. In particular, the authors give the sufficient condition for the well-posedness of the fixed point problem under any norms, and provide the corresponding contraction factor. Consequently, the author derives the Lipschiz constant of their implicit model induced by the infinity norm and use the model for robustness tasks.

**Limitations And Societal Impact:**

Yes.

**Main Review:**

# Strength
This paper studies an important and interesting area. Implicit modelling starts to gain more attention as it's a good method to save memory while achieving comparable performance. The particular problem tackled by this paper is also important as previous results mostly focus on l2 norms and it is important to extend the existing results to other norms like l_infinity norm. The derivation is non-trivial. Although I didn't check the proof detail, the claims do seem sound and the paper is written clearly.

# Weakness
- In terms of writing, this paper has several big chunks of theorems but lack sufficient discussion of their significance. For example, the osLip is a weaker notion than the Lip constant, I think it's helpful that the author can provide some example where these two notions differ by a lot.
- A comparison of the matrix measure and largest eigenvalue may also be helpful. For example, in corollary 5, the sufficient condition for the uniqueness of fixed point is that u(A)<1. I think this coincide with the largest eigenvalue of A if the underlying norm is l_2 (correct me if I'm wrong). It is valuable to include such comparisons in the discussion. I do realize that the authors have discussed about the PF eigenvalue used in El Ghaoui et al., I would also like to see a comparison to the condition used in Winston and Kolter.
- The training problem is framed as a constrained optimization problem. To me this is not a desired property. I would guess that at every gradient step we have to project the updated parameters on the constrained set and it can be very time/resource consuming. It would be great if there's a way to re-parameterize the weights as done in Winston and Kolter.
- More training details can be added. For example, the dimensions of A, B, C, D are missing in section 6. I would also want to see two more results: how many iterations does it take for forward and backward damped iterations, and what's the wall-clock time to train these models.



**Time Spent Reviewing:**

3

---

> ### Author Response · Authors · 2021-08-10
> **Response to Reviewer ivrZ**
>
> *We would like to thank this reviewer for their kind review and thoughtful feedback regarding our manuscript.*
> ___
>
> *"In terms of writing, this paper has several big chunks of theorems but lack sufficient discussion of their significance ..."*
>
> Thank you for your comment. These issues will be addressed in the final version by adding discussions after each theorem/corollary highlighting its importance and providing insights into its applications. Regarding the comparison between osL and Lip, we would like to refer the reviewer to Appendix A, where osL and Lip of a special class of $2\times 2$ matrices are illustrated in Figure 4. For a simple non-linear scalar example, consider the function $f(x) = -x^3$. This function is not globally Lipschitz (i.e, $\mathrm{Lip}(f) = \infty$) but it is one-sided Lipschitz with $\mathrm{osL}(f) = 0$. We thank this reviewer for this suggestion and we will add more details including the above simple example to the final version of the paper.
> ___
>
> *"A comparison of the matrix measure and largest eigenvalue may also be helpful ..."*
>
> This is a great suggestion. First we should mention that the notion of largest eigenvalue of a matrix is ambiguous unless the matrix is symmetric. Thus, we instead use the notion of "spectral abscissa'' (i.e., max real part of eigenvalues of the matrix) and focus on comparing the matrix measure of $A$ with this notion. The connection between matrix measures and spectral abscissa is interesting and well-studied in the literature. We have a sentence in Section 2, mentioning that ``the matrix measure is a tighter upper bound on the spectral abscissa of $A$ than $||A||$''. Also, property (21-d) in Appendix A provides upper and lower bounds on the eigenvalues of the matrix based on its matrix measure. Regarding $\mu_2(A)$, for a non-symmetric matrix $A$, it is not generally true that $\mu_2(A)$ coincides with the spectral abscissa of the matrix $A$.
>
>
> We thank this reviewer for pointing to this interesting and important comparison of our well-posedness condition with MON in [2]. We have drawn an analogy between our approach with MON [2] and implicit deep learning [1] in Appendix D.1. However, we agree with this reviewer that this comparison is essential in understanding the connection with the literature. Therefore, space permitting, we will move this discussion to the main body of the paper. Regarding the MON well-posedness condition obtained in [2], we have shown in Appendix D.1 that it is equivalent to $\mu_2(A)<1$. However, we would like to mention that this condition is not equivalent to our well-posedness condition (i.e., $\mu_{\infty,[\eta]^{-1}}(A)<1$). Indeed there are classes of matrices which satisfy our well-posedness condition while they do not satisfy $\mu_2(A)<1$ and vice versa. We will add a picture in the final version to show this fact for certain classes of $2\times 2$ matrices. We would be glad to provide this picture and more details on this entire discussion if there is an approved manner to share blind documents from authors to reviewers of NeurIPS 2021.
> ___
>
> *"The training problem is framed as a constrained optimization problem. To me this is not a desired property ..."*
>
> This is a great point! Our Lemma 9 in Appendix B presents a parametrization of matrices with bounded $\ell_\infty$-norm matrix measure. This parametrization allows us to circumvent the $\mu_\infty$ constraint in the training problem and obtain an unconstrained optimization problem. We have applied this parametrization in the latest version of our code and we can confirm that incorporating this parametrization in our code leads to a shorter runtime with a very comparable performance for training. We agree that we should have highlighted the role of this parametrization in our manuscript. In the final version, we will add a discussion about this parametrization and its role in the training of our implicit neural network in Section 5 of the paper.
> ___
>
> *"More training details can be added. For example, the dimensions of $A$, $B$, $C$, $D$ ..."*
>
> In the final version of the paper, we will add the dimensions of matrices $A$, $B$, $C$, $D$ in Section 6.
>
> Regarding the number of iterations and the wall-clock time to train our model, we refer to our top comment regarding "**Runtime and number of iterations.**"

---

> ### Comment · Area_Chair_jcVp · 2021-08-21
> **Does the author response address your concerns?**
>
> Dear Reviewer,
> I would encourage you to read the author rebuttal and give your further comments. Please pay attention to whether you want the authors to read your further comments. Thanks!
>
> Area Chair

---

> > ### Comment · Reviewer_ivrZ · 2021-09-02
> > **Thanks for the response**
> >
> > Thank the authors for their responses, my concerns are addressed and I think this is a good paper for the implicit modeling field. Hence I'm raising my score to 7.

---

### Official Review · Reviewer_r3Ki · 2021-07-15

**Rating:** 6
**Confidence:** 3

**Summary:**

The paper proposes a new well-posedness condition for DEQs that is less conservative than the l_\infty norm proposed in [El Ghaoui et al., 2019]. A Lipschitz regularization is also developed to improve the robustness to some classes of adversarial perturbations. Several experiments show the effectiveness of the proposed method.
___________________________________________________
The author responses address most of my concerns. So I raised the score and tended to acceptance.

**Limitations And Societal Impact:**

I have two main concerns.
First, how to choose \gamma in (16) for a DEQ training problem? Also, it is better to study the effect of different settings of \gamma. Why set \gamma = 0.5 in subsection 6.1 and \gamma=0.95 in subsection 6.2?

Second, it is better to plot the running time vs. test accuracy, e.g., in Figure 2. Does the proposed method outperform MON and l_\infty norm in training efficiency?

**Main Review:**

Studying the well-posedness and convergence of the fixed-point equations of DEQs is very important. Based on contraction theory for the l_\infty norm, the paper develops a new well-posedness condition for DEQs. The paper is well-written and organized. However, the advantage over MON is not clear. Note that MON does not consider the robustness.

**Time Spent Reviewing:**

5

---

> ### Author Response · Authors · 2021-08-10
> **Response to Reviewer r3Ki**
>
> *Thank you for carefully reading our manuscript and for your comments*.
>
> ___
>
> *"The paper is well-written and organized. However, the advantage over MON is not clear. Note that MON does not consider the robustness."*
>
> Thank you for this important comment. We believe that our approach has several advantages over the framework of MON/LBEN which is detailed in our top comment regarding "**More comprehensive comparison with MON in [2] and LBEN in [3].**"
>
> ___
>
> *"First, how to choose $\gamma$ in (16) for a DEQ training problem? Also, it is better to study the effect of different settings of $\gamma$. Why set $\gamma = 0.5$ in subsection 6.1 and $\gamma=0.95$ in subsection 6.2?"*
>
> For $\gamma \in [0,0.95]$ we did not observe significant change in our accuracy over the MNIST dataset. However, the choice of $\gamma$ affects the Lipschitz constant of the implicit network and thus affects the robustness of the network as is shown in Appendix D.
>
> Regarding the use of $\gamma=0.5$ in the simulations of Section 6.1, since the reference [1] chose the value $\gamma=0.5$ for its simulations, we decided to pick $\gamma=0.5$ to be able to compare our results with the simulations in [1]. However, in the final version of the paper, based on the suggestions of other reviewers, we will remove Section 6.1 and the numerical simulation for learning the scalar function and instead will add the simulations for implicit convolutional neural networks on CIFAR-10. Regarding the choice of $\gamma$ for CIFAR-10 dataset, we pick values of $\gamma$ which are consistent with the monotonicity parameter $m$ of MON in [2].
> ___
>
> *"Second, it is better to plot the running time vs. test accuracy, e.g., in Figure 2. Does the proposed method outperform MON and $\ell_\infty$ norm in training efficiency?"*
>
> Thank you for your suggestion. We agree that comparing the experimental results for the runtime of the training algorithm is of significance. We did not plot running time vs. test accuracy since this result is not reported neither for the implicit deep learning approach in [1], nor for MON in [2] and nor for LBEN in [3]. All of these papers focus on presenting the experimental results in the form of test accuracy vs epochs. In the final version of the paper, we will include the time required per epoch for a comparison with the implicit deep learning in [1] and MON in [2].
>
> It should be noted that in this paper, we propose an averaged iteration to compute forward and backward passes, compared to MON which uses the Peaceman-Rachford algorithm and the implicit deep learning model in [1] which uses Picard iterations. Compared to the implicit deep learning model in [1], our training process is slower as the averaged iterations allow us to compute the equilibrium for a larger class of matrices at the expense of a slower rate of convergence. This larger class of matrices allows us to improve the performance and robustness of our models. We refer this reviewer to our top comment regarding "**Runtime and number of iterations**" for a more detailed comparison with MON.

---

> ### Comment · Area_Chair_jcVp · 2021-08-21
> **Does the author response address your concerns?**
>
> Dear Reviewer,
> I would encourage you to read the author rebuttal and give your further comments. Please pay attention to whether you want the authors to read your further comments. Thanks!
>
> Area Chair

---

### Official Review · Reviewer_Q7s2 · 2021-07-18

**Rating:** 6
**Confidence:** 4

**Summary:**

This paper presents a thorough analysis of constructing provably convergent equilibrium networks in the non-Euclidean space. Instead of directly requiring Lipschitzness as in prior works (which mostly dealt with $L_2$), the authors show that we can further relax the constraint by considering one-sided Lipschitzness and exploit the contraction theory via semi inner products. The experimental results of the paper seem to demonstrate the validity of the theoretical results and the paper provides empirical evidence on the potential application of the theories in improving implicit networks' adversarial robustness.

**Limitations And Societal Impact:**

This is a theoretical work and I think the limitations are well-addressed (mainly in the form of assumptions and sufficiency analysis in theorems).

**Main Review:**

Overall, I enjoy reading this paper, which provides a valuable discussion on how to extend (and exploit) implicit networks' stability analysis to non-Euclidean spaces. I believe this paper will be of interest to the relevant audience in the NeurIPS community and extends the current efforts (as pointed by the authors in the paper) to build provably stable implicit deep networks and scale them to more general settings. At the same time, while the final application of the theories (e.g., Thm 1 and 2) are for the equilibrium networks, the techniques used in this paper and part of the conclusions have been built on top of prior works that studies; e.g., [1,2,3]. Moreover, I find some critical empirical analysis missing. In summary, I find the following strengths and weaknesses:

Strengths:
- Clear explanation of the theory. I checked most of the proof (mainly appendix A & C) and they are well-written and correct. The theoretical motivation for using osL-ness and damped fixed point iteration is well explained and quite smart.
- The paper extends prior works to non-Euclidean space and presents a new framework/condition through which we can build well-posed implicit networks (i.e., practical significance).

Weaknesses:
- Limited theoretical significance (to clarify, the application of such kind of contraction analysis to implicit neural networks is novel, and probably significant; but not the osL-based contraction theory itself).
- Missing experimental details and comparisons (e.g., with Revay et al., which dealt with $L_2$; and on a wider range of experiments). The comparison with El Ghaoui is good but it's still only a weak baseline.
- Still relatively weak empirical evidence on the benefit of applying the analysis in practice (i.e., having this larger parameter search space). Specifically, the advantage of using the proposed $L_\infty$-based and relaxed-constraint formulation over MON or even the LBEN model in Revay et al. is not clear to me when in the presence of strong adversarial attacks, such as PGDM and FGSM (see Appendix D).

-------------------------------------------------------------

More questions/comments:

1. One particular theoretical benefit of the approach is the use of osL analysis and measure $\mu_{\infty [\eta]^{-1}}(A)$ rather than the simplest matrix norm. This begs the question of how the pick of $\eta$ affects the empirical performance (theoretically it's only a matter of existence issue, but I'm more interested in practical implications). The authors picked $\mathbf{1} \in \mathbb{R}^n$ here. Could the authors expand more on this? Moreover, in the submitted code "trainMNISTmodel.py", I did not find any mechanism for actually ensuring $\mu_{\infty, [\eta]^{-1}}(A) < \gamma$ (you did compute the LHS, which is
```python
X = torch.abs(model.A) - torch.diag(torch.diag(torch.abs(model.A))) + torch.diag(torch.diag(model.A))
matmeas = torch.max(X.sum(1))
```
but didn't quite use it; but it's likely that I missed something). How is the diagonally weighted measure preserved during training?

2. One question that I found to be absent in the empirical discussion is the convergence rate. Although damped fixed point iteration helps us relax the Lipschitz condition a bit, it is also known to possibly bring the extra risk of slower fixed-point convergence than the non-average iterations. In the case where $\alpha$ is small, I wonder how efficient this (albeit provable) convergence is?

3. You may want to cite [4] as well, which is clearly related (but in Euclidean space).

4. It seems to me that the results of Sec. 4 should be able to generalize even further to the case where $Ax$ is replaced by a non-linear transformation $\Lambda(x)$, where $\Lambda$ is everywhere differentiable. In that case, you only need to replace the $A$-based results in the conclusion with $D\Lambda$. Is there any reason for not doing so in theory and in practice? E.g., the matrix measure of $D\Lambda$ would be hard to compute?


[1] https://arxiv.org/pdf/2103.12263.pdf
[2] https://hal.archives-ouvertes.fr/hal-01313105/document
[3] https://link.springer.com/content/pdf/10.1007/978-3-540-72234-2.pdf
[4] https://openreview.net/pdf?id=VcB4QkSfyO

-----------
-----------

#### Post-rebuttal

I appreciate the authors' response. Overall, while I agree this paper has presented a solid work, I personally would love to see larger-scale, more practical, and more performant results of these provable implicit models and what we could achieve with them.

**Time Spent Reviewing:**

6.5

---

> ### Author Response · Authors · 2021-08-10
> **Response to Reviewer Q7s2**
>
> *We would like to thank this reviewer for carefully reading our manuscript and code and providing helpful comments and insightful suggestions.*
>
> Please refer to our comments at the top regarding (i) the theoretical significance of our approach outside of implicit neural networks, (ii) advantages of our model over MON/LBEN, and (iii) missing experimental details and comparisons.
> ___
> *"Limited theoretical significance (to clarify, the application of such kind of contraction analysis to implicit neural networks is novel, and probably significant; but not the osL-based contraction theory itself)."*
>
> We agree that there is a rich literature on contraction theory going back to [9]. However, we respectfully disagree that our theoretical results have limited significance. We agree that the treatment in Appendix A is not novel and is built upon the '98 work of Lohmiller and Slotine and other papers in the contraction literature as well as the recent manuscript [10]. However, we believe that our paper's theoretical contributions go well beyond restating the known results from the literature. We refer this reviewer to our top comment regarding "**Theoretical significance outside of implicit neural networks**" for a complete list of our theoretical contributions.
> ___
> *"Missing experimental details and comparisons (e.g., with Revay et al., which dealt with $\ell_2$; and on a wider range of experiments). The comparison with El Ghaoui is good but it's still only a weak baseline."*
>
>  Regarding the comparison with LBEN in [3], unfortunately, we could not find the LBEN code neither on Github nor on the any of the authors’ websites and it seems that the code for LBEN is not publicly available. Regarding a wider range of experiments, we refer this reviewer to our top comment regarding "**More experiments**".
>
> Regarding comparison with the literature, apart from comparison with the implicit deep learning framework of El Ghaoui et al. in [1], we have also compared the accuracy and robustness of our implicit neural network with respect to the MON framework in [2], which appears to be the state-of-the-art framework for implicit neural networks (see Section 6.2 and Appendix D).
> ___
>
> *"Still relatively weak empirical evidence on the benefit of applying the analysis in practice ..."*
>
> We believe that our approach has several advantages over the framework of MON/LBEN which we address in the top comment regarding "**More comprehensive comparison with MON in [2] and LBEN in [3].**"
>
> Regarding the advantages of our framework in presence of strong adversarial attacks, we note that both PGDM and FGSM attacks are designed using the $\ell_\infty$-norm perturbations of the input (see equations (43) and (44)) and thus it is more natural to analyze their effect on the neural network using our $\ell_\infty$-based framework than the $\ell_2$-norm based frameworks of MON and LBEN.
> ___
> *"One particular theoretical benefit of the approach is the use of osL analysis and matrix measure rather than ..."*
>
> This is a great question! The choice of diagonal weights $\eta$ provides our framework with additional flexibility in choice of norm and allows for a larger class of matrices to satisfy our well-posedness condition. However, at this time, our preliminary numerical experiments do not show any significant change in performance due to the choice of $\eta$. In our experiments in Section 6, we picked $\eta = \mathbb{1}_n$ primarily for a more direct comparison with the MON models [2] and implicit deep learning model [1] where their $\ell_2$- and $\ell_\infty$- norms, respectively, are not diagonally weighted. In future work, we intend to investigate algorithmic methods to design $\eta$.
>
> Regarding the code, in its original version, we were performing a projected gradient descent to enforce the osL constraint (it is in the forward method of the ContractingFunctionInf class contained in ContractingFunction.py). The code you have highlighted was part of a sanity check to ensure that the projection was working. In the latest version of the code, we have implemented the parametrization in Lemma 9 in Appendix B to directly enforce the osL constraint and remove the projection.
> ___
>
> *"One question that I found to be absent in the empirical discussion is the convergence rate ..."*
>
> We agree with this reviewer that in the current form our training can allow for very small step sizes and this may lead to slow convergence to the solution of the fixed-point problem and thus slow training. However, empirically, this situation never happened in our simulations of Sections 6.1 (for function estimation) and in Section 6.2 (for the MNIST dataset) as well as for our recent simulations on CIFAR-10. To further remedy this issue with small step-sizes, we additionally implemented an adaptive step-size scheme to optimize the rate of convergence to the fixed-point which alleviates this issue.
> ___
>
> *"You may want to cite [4] as well, which is clearly related (but in Euclidean space)."*
>
> Thank you for the suggestion. We agree that [4] is an important reference that we have missed. In the final manuscript, we will add a comparison between the approach in [4] and our framework which we summarize here:
>
> 1) our framework can study well-posedness and estimate the input-output Lipschitz bounds for implicit neural networks with a larger class of activation functions. While [4] only focuses on implicit networks with ReLU activation functions, our framework is applicable to implicit neural networks with weakly increasing non-expansive activation functions.
> 2) the approach in [4] only provides input-output Lipschitz bounds with respect to the $\ell_2$-norm (see Theorem 1 in [4]) while in Corollary 5(iii) we provide input-output Lipschitz bounds of the neural networks with respect to arbitrary norms on their input and output spaces and  $\ell_\infty$-norm for the hidden layers.
> ___
>
> *"It seems to me that the results of Sec. 4 should be able to generalize even further ..."*
>
> This is a great question! Of course you are completely correct in that we can further generalize to nonlinear differentiable transformations, where the well-posedness condition is instead $\mu_{\infty,[\eta]^{-1}}(D\Lambda) < 1$. In this setting, several interesting questions arise including how to efficiently parametrize $\Lambda$ so that this constraint is always enforced. We believe this is a very interesting generalization and could be a topic of future investigation. In this paper, we confine ourselves to the standard model with $Ax$ for the sake of simplicity and to get a fair comparison with implicit deep learning model in [1] and MON in [2].

---

> ### Comment · Area_Chair_jcVp · 2021-08-21
> **Does the author response address your concerns?**
>
> Dear Reviewer,
> I would encourage you to read the author rebuttal and give your further comments. Please pay attention to whether you want the authors to read your further comments. Thanks!
>
> Area Chair

---

### Official Review · Reviewer_xt3G · 2021-07-22

**Rating:** 7
**Confidence:** 4

**Summary:**

This paper gives a novel characterization of well-posedness for implicit networks based on the one-sided Lipschitz constant (osL). This follows in a line of work of other such characterizations [1,2,3,4], but differs in that well-posedness is guaranteed by a constraint on the osL of the weights matrix, which can be wrt arbitrary norms. The authors focus on the $\ell_\infty$ norm, for which they show accelerated convergence of dampened fixed-point iterations.  They also drive a bound for the Lipschitz constant of the network wrt the $\ell_\infty$ norm on the input space, which is more natural for achieving robustness than the $\ell_2$-based Lipschitz bounds of [3] and [4]. They train networks while enforcing well-posedness and using the Lipschitz bound as a regularizer, and demonstrate the empirical robustness of the models.

**Limitations And Societal Impact:**

* The authors state that the work is fundamental research so will not lead to negative societal impact.
* In terms of limitations of the work, it would be great if the authors could report runtime and number of function evals for their models, which are typically quite high for such models.

**Main Review:**

Overall, I think the connection between non-Euclidian contraction theory and implicit networks is quite interesting and will be beneficial for people working in this area. However, there are a number of comparisons to existing works that are lacking, both in the theory and experiments (see below). Additionally, some of the benefits described in the theory are not well demonstrated by the experiments.

This paper addresses an important topic, since implicit networks are growing in practicality and popularity, so understanding and improving their robustness properties is important. If the authors could address some of my concerns, I'd be inclined to increase my score.

### Pros:

*  The application of non-Euclidean contraction theory to implicit networks appears to be novel and provides a nice way to guarantee contraction wrt the $\ell_\infty$ norm and well-posedness.
* The accelerated convergence in the $\ell_\infty$ norm seems important, since improving convergence times has been a practical consideration for implicit networks.
* It's great to see that the Lipschitz bound can be computed for arbitrary norms via Lim's lemma, which is more general and elegant than the $\ell_2$ bounds in [4].
* Bounding the Lipschitz constant wrt the $\ell_\infty$ norm is valuable since it leads to robustness to $\ell_\infty$ attacks, which are arguably more natural than $\ell_2$ attacks.

###  Cons:
* I think the paper would be much improved by better comparisons to exiting work. In particular, [4] also derives Lipschitz bounds for MON but is not cited.  It looks like Theorem 3 generalizes the bound in [4], while being a more elegant way of deriving it.
* It would be good to understand how the set of networks characterized here compares to that of [2] and [3]. E.g. along the lines of Figure 2 in [3], which does a nice job of comparing the constraint in [2] and [3], showing how [3] is less restrictive. Do the osL-bounded networks correspond exactly to the set of MON networks in [2] if using $\ell_2$ instead of $\ell_\infty$ norm? I believe this is stated in Appendix D but could be elaborated on in the main text.
* I don't know how much of the theory is novel outside of the implicit networks literature. Could the authors comment on how Theorem 3 compares to the results in [5]?
* It is unclear how the osL constraint is enforced during training. In Lemma 9, appendix B, the authors provide a parameterization of the well-posed network like those in [2] and [3], which allows for unconstrained training. But that doesn't seem to be used during training here?
* Experimental comparison to LBEN [3] is missing, and it is unclear what value of monotonicity parameter is used for MON [2]. As is shown in [4], this parameter controls the robustness (at least to $\ell_2$ attacks).
* It would be nice to see certified robustness, since the authors provide a concrete bound on the $\ell_\infty$ norm.
* It would be nice to see results on CIFAR-10, since robustness on MNIST is not as meaningful.
* It's unclear what the take-away from the first experimental section is meant to be.
* It would seem important to show if the accelerated convergence can be realized experimentally, and compare convergence to that of LBEN/MON.

### References:
* [1] [El Ghaoui et al. 2019. *Implicit deep learning*](https://arxiv.org/pdf/1908.06315.pdf)
* [2] [Winston and Kolter 2020. *Monotone operator equilibrium networks*](https://arxiv.org/pdf/2006.08591.pdf)
* [3] [Revay et al. 2020. *Lipschitz Bounded Equilibrium Networks*](https://arxiv.org/pdf/2010.01732.pdf)
* [4] [Pabbaraju et al. 2020. *Estimating Lipschitz constants of monotone deep equilibrium models*](https://openreview.net/forum?id=VcB4QkSfyO)
* [5] [Adly et al. 2014 *On one-sided Lipschitz stability of set-valued contractions*](https://06658af9-a-62cb3a1a-s-sites.googlegroups.com/site/adontchev/lim.pdf)

**Time Spent Reviewing:**

5

---

> ### Author Response · Authors · 2021-08-10
> **Response to Reviewer xt3G**
>
> *We would like to thank this reviewer for carefully reading our manuscript and providing helpful comments and suggestions.*
>
> Please refer to our comments at the top regarding (i) the novelty of our method outside of implicit neural networks, (ii) more experiments on CIFAR-10, and (iii) runtime and number of iterations.
> ___
>
> *"I think the paper would be much improved by better comparisons to exiting work. In particular, [4] ..."*
>
> Thank you for the suggestion. We agree that [4] is an important reference that we missed. In the final manuscript, we will add the following comparison between the approach in [4] and our framework:
> 1) our framework establishes well-posedness and input-output Lipschitz bounds for implicit neural networks with a larger class of activation functions than [4]; indeed [4] only considers the ReLU activation functions but our framework uses the class of weakly increasing non-expansive activation functions,
> 2) [4] provides input-output Lipschitz bounds with respect to the $\ell_2$-norm (see Theorem 1 in [4]). Instead, our Corollary 5(iii) provides input-output Lipschitz bounds with respect to arbitrary norms on the input and output spaces and $\ell_\infty$-norm for the hidden layers.
>
> Thank you for you comment regarding our Theorem 3. We'd like to add that our Theorem 3 applies to any arbitrary map $F$ with $\mathrm{osL}(F)<1$ and thus it is more general than Theorem 1 in [4] which is applicable only to implicit neural network of the form given in equation (4) in [4].
> ___
> *"It would be good to understand how the set of networks characterized here compares to that of [2] and [3]..."*
>
> Thank you for your careful comment and great suggestion. As we mentioned in Appendix D.1 the well-posedness condition in [2] is equivalent to $\mu_2(A)<1$ and the well-posedness condition in [3] is equivalent to $\mu_{2,[\eta]^{-1}}(A)<1$ for some positive vector $\eta$. On the other hand, our well-posedness condition is of the form $\mu_{\infty,[\eta]^{-1}}(A)<1$. One can show that the sets of matrices $A$ satisfying $\mu_{2,[\eta]^{-1}}(A)<1$ and the set of matrices $A$ satisfying $\mu_{\infty,[\eta]^{-1}}(A)<1$ are overlapping but distinct (neither a subset of the other). We will add a picture in the final version to show this fact for certain classes of $2\times 2$ matrices.
>
> Regarding the comparison with MON in [2],
> 1) our implicit neural networks have the same architecture as MON,
> 2) our implicit neural networks have similar (but distinct) well-posedness conditions as MON (our well-posedness condition is of the form $\mu_{\infty,[\eta]^{-1}}(A)<1$ while for MON the well-posedness is of the form $\mu_2(A)<1$),
> 3) our framework proposes a different parametrization of feasible sets (our parametrization is given in Lemma 9 while for MON the parametrization is given in [Proposition 1, 2]),
> 4) our framework has a conceptually simpler and computationally more efficient treatment of activation functions; we consider the class of non-increasing and non-expansive activation functions while MON considers the class of activation functions representable as proximal operators of closed convex proper maps. We note that in the MON framework finding the associate map of the proximal operator may introduce undesirable computational complexity in the iterations for computing the network fixed-points as well as the backpropagation.
>
> We agree with this reviewer that this comparison with references [2] and [3] is important and thus in the final version of the paper, we will add a discussion about it and move some of these material from Appendix D to the main body of the paper.
> ___
> *"Could the authors comment on how Theorem 3 compares to the results in [5]?"*
>
> Regarding Theorem 3 and its connection with [5], we believe paper [5] deals with a different concept of “one-sided (outer/inner) Lipschitz stability”, which is applicable to multi-valued maps in arbitrary metric spaces. Roughly speaking, in the definition of Lipschitz multi-valued map the symmetric Hausdorff distance is replaced by an asymmetric “distance-like” metric.
> ___
> *"It is unclear how the osL constraint is enforced during training. In Lemma 9, Appendix B,..."*
>
> It is true that, simply due to the chronology of results, the parametrization in Lemma 9 has not been utilized in the training simulations. In the previous version of the code, we were applying a projected gradient descent to enforce the osL constraint (it is in the forward method of the ContractingFunctionInf class contained in ContractingFunction.py). Since the initial submission, we have updated the code to use the parametrization in Lemma 9. In the final version of the paper, we will move the parametrization into the main body, explain its importance in training, and report numerical results only for the parametrization-based unconstrained training. As a preview, our new simulations confirm that incorporating this parametrization in our code leads to (1) shorter runtime and (2) very comparable neural network performance.
> ___
> *"Experimental comparison to LBEN [3] is missing, and it is unclear what value of monotonicity parameter ..."*
>
>  Regarding experimental comparison with LBEN in [3], unfortunately, we could not find the code for LBEN in [3] neither on Github nor on the any of the authors’ websites and it seems that the code for LBEN is not publicly available. Regarding our simulations of MON in [2], based on our Appendix D.1, the monotonicity parameter $m$ used in [2] (see Proposition 1 and equation (5) there) corresponded to $1-\gamma$ in our framework. As a result, specifying the value of $\gamma$ for MON (as in $\mu_2(A)\le \gamma$ or $\tfrac{1}{2}(A+A^T)\preceq \gamma I$) completely characterizes its monotonicity parameter. Note that the value of the parameter $\gamma$ is always specified in our numerical experiments. For instance see Sections 6.1 and 6.2 and Figures 7, 9, 11, and 12 in Appendix D.
> ___
>
> *"It would be nice to see certified robustness, since the authors provide a concrete bound on the $\ell_{\infty}$ norm"*.
>
> You make a good point! Following the approach in [4], we will compute and generate a plot of the certified robustness of our models (similar to Figure 4(a) in [4]) and add it to the final version of our paper.
>
> ___
>
> *"It's unclear what the take-away from the first experimental section is meant to be."*
>
> We included the function estimation experiment in Section 6.1 to present solution of a "regression" learning problem using implicit networks whereas the experiment in Section 6.2 for the MNIST dataset presents the solution of a "classification" learning problem. However, we agree with this reviewer that the learning experiment in Section 6.1 is elementary and only serves as an academic example. In the final version of the paper we remove this Section and replace it with the experiment on CIFAR-10.
> ___
> *"It would seem important to show if the accelerated convergence can be realized experimentally, and compare convergence to that of LBEN/MON."*
>
> The accelerated convergence is indeed realized in experiments. Theorem 2 shows that the averaged iterations enjoy an accelerated convergence rate for diagonally weighted $\ell_1$-norms and $\ell_\infty$-norms. In all our numerical experiments in Section 6, we pick the step-size for the averaged iterations to be larger than or equal to the upper bound for the step-size given in Theorem 2(i). In our code, to further accelerate our algorithm, we additionally implement an adaptive step-size scheme to optimize the rate of convergence to the fixed-point. Regarding the comparison with LBEN/MON, one should note that in LBEN/MON the Peaceman-Rachford splitting algorithm is used to speedup the convergence of the fixed-point problem. The number of iterations and the runtime of these two algorithms are reported in the top comment.

---

> > ### Comment · Reviewer_xt3G · 2021-08-28
> > **Thank you for your reply**
> >
> > Thanks very much for your detailed reply, which addresses the majority of my concerns. I'm raising my score to a 7, based on the following improvements:
> >  *  Additional experiments with CNNs on CIFAR-10
> >  * More thorough comparison to existing parameterizations
> >  * Analysis of convergence
> >  * Unconstrained training using the parameterization
> >
> > I do have a few follow up questions:
> >
> > _"Could the authors comment on how Theorem 3 compares to the results in [5]?"_
> >
> > I agree that [5] deals with a different concept of osL, but my question was whether [5] is a generalization of your's, and whether Theorems 3 and 4 are generalizations of your Theorem 3. (Admittedly I haven't fully grasped [5] and could be missing something.) However, since you are only claiming Theorem 3 as a "natural extension of Lim's Lemma" and not a major contribution, this is not too important.
> >
> > ---
> >
> > _"It would seem important to show if the accelerated convergence can be realized experimentally, and compare convergence to that of LBEN/MON."_
> >
> > I'm glad to see the new  iteration count and training time results. I'd be interested to additionally see both Peaceman-Rachford and averaged iterations for both your model and MON reported on comparable _fully-trained_ models, since it increases over the course of training. However, this won't affect my score.
> >
> > ---
> >
> > _"Experimental comparison to LBEN [3] is missing, and it is unclear what value of monotonicity parameter ..."_
> >
> > Got it. However, it was not clear to me from the text in 6.1 that $\gamma$ of 0.95 also applied to the MON.  I do see it mentioned in the appendix. The reason I was curious is because [4] indicates that changing $\gamma$ changes the $\ell_2$ robustness, so you may consider comparing to the MON with multiple $\gamma$ values.

---

> > > ### Author Response · Authors · 2021-08-31
> > > **Responses to follow-up questions**
> > >
> > > Thank you for your positive evaluation of our new changes. We address your follow-up questions below:
> > > ___
> > >
> > > "*I agree that [5] deals with a different concept of osL, but my question was ...*"
> > >
> > > Thanks for your great question. We believe that the notion of one-sided (outer/inner) Lipschitz constant introduced in [5] is different (and not an extension) of our notion of one-sided Lipschitz constant. Roughly speaking, reference [5] deals with fixed-points of set-valued maps and the notion of Lipschitz (outer/inner) continuity introduced in [5] is a set-valued notion which, for single-valued maps, coincides with the classical Lipschitz continuity. On the other hand, our notion of one-sided Lipschitz continuity is weaker than Lipschitz continuity. Therefore, we believe that Theorems 3 and 4 in reference [5] are generalization of the classical Lim's lemma to multi-valued maps but they are not extensions of our Theorem 3.
> > > ___
> > >
> > > "*I'm glad to see the new iteration count and training time results. I'd be interested to...*"
> > >
> > > That is a good point! In the final version of the paper, we will report the iteration count and evaluation time for both Peaceman-Rachford and averaged iterations on the test-data.
> > > ___
> > > "*Got it. However, it was not clear to me from the text in 6.1 that ...*"
> > >
> > > In the final version of the paper, we will add the exact value of $\gamma$ for MON in Section 6.1. You make a good observation regarding the connection between $\gamma$ and the robustness of MON (as is shown in [4]). In the final version of the paper, we will plot accuracy vs. $\ell_\infty$ perturbations for our INN model as well as MON with several values of $\gamma$.

---

> ### Comment · Area_Chair_jcVp · 2021-08-21
> **Does the author response address your concerns?**
>
> Dear Reviewer,
> I would encourage you to read the author rebuttal and give your further comments. Please pay attention to whether you want the authors to read your further comments. Thanks!
>
> Area Chair

---

### Author Response · Authors · 2021-08-10
**Response to common reviewer comments**

*We would like to thank the reviewers for their detailed reading and careful insights and comments on our manuscript. We would like to address a few of the common comments here. We have also answered the specific questions in the individual replies.*
___

 **More comprehensive comparison with MON in [2] and LBEN in [3]**: We believe that our approach has several advantages over the framework of MON/LBEN which we summarize below:
1) the MON/LBEN implicit neural networks adopt as activation functions the proximal operators of closed convex proper maps; instead our INN adopts weakly increasing non-expansive activation functions. While the connection between these two classes of activation functions is not fully established in the literature, we believe that our framework leads to a more natural treatment of activation functions and avoids the undesirable computational complexity associated with finding and computing the suitable proximal operator in both the fixed-point iterations and backpropagation.
2) note that the follow-up paper [4] studies the robustness and input-output Lipschitz bounds of MON. However, the analysis in MON/LBEN and also the analysis in [4] are restricted to $\ell_2$ input-output Lipschitz bounds. We present a framework for the systematic analysis of the $\ell_\infty$ input-output Lipschitz constants of implicit neural networks. We believe that the $\ell_\infty$ input-output Lipschitz bounds of neural network are particularly important in applications where understanding the propagation of the worst-case entry-wise error through the neural network is more relevant than understanding how the $\ell_2$ norm of the error propagates through the network.
3) Empirically, at the price of losing few percentage points of clean performance accuracy, our framework can obtain large robustness improvement with respect to sizeable $\ell_\infty$- perturbations. This is visible in our empirical robustness analysis for the MNIST dataset. For instance, for the continuous image inversion attack shown in Figure 3 (right image) for perturbations with $\ell_{\infty}$-norm equal to 0.2, our implicit model with the regularization parameter $\lambda=10^{-2}$ obtains around 78% accuracy while the MON model as well as the implicit deep learning model in [1] obtain around 18% accuracy. Similarly, for PGDM attacks, Figure 11 shows that for the perturbations with $\ell_\infty$-norm equal to 0.05, our implicit model with regularization parameter $\lambda=10^{-2.5}$ achieves around 72% accuracy while the MON and implicit deep learning in [1] achieve around 35% accuracy.

**Theoretical significance outside of implicit neural networks**: We believe our paper includes the following novel theoretical results of interest even outside the implicit networks literature:
1) Theorem 1 is novel. Specific results for the Euclidean norm exist in the classical literature on monotone operator theory (see [6]), but our result is new for general norms.
2) Theorem 2 is novel and it is based on Lemma 7 which is also novel. Indeed, we believe that Lemma 7 is the first result to show this affine relationship between matrix measures and matrix norms for diagonally weighted $\ell_1$ and $\ell_\infty$ norms.
3) Theorem 3 is novel as stated, but it is a natural extension of an old result by Lim (see [8]).
4) Lemma 8 provides novel matrix measure results whose careful comparison with the literature is given in Appendix B.

While Theorem 4 is specifically stated for implicit neural networks, Theorems 1, 2, and 3 are stated for a general map $F$ with $\mathrm{osL}(F)<1$ and thus are applicable to a large class of problems outside implicit neural networks. Therefore, we believe that Theorems 1-3 can be considered as a first step toward developing a non-Euclidean monotone operator theory akin to the classical monotone operator theory (see for example [6] and [7]). Lemmas 7 and 8 are applicable to general matrices that may appear in a wide range of problems outside neural networks.

**More experiments**: Based on suggestions of several reviewers, we will add experiments on the CIFAR-10 dataset. The MON approach in [2] defines suitable implicit CNNs to perform experiments on CIFAR-10. After the initial submission, we realized that we can extend our framework and define implicit CNNs with non-Euclidean well-posedness conditions. Here is a brief summary: We let $u\in \mathbb{R}^{d s^2}$ be a $d$-channel input of size $s\times s$ and $x\in \mathbb{R}^{n s^2}$ be a $n$-channel hidden layer. To define our implicit CNN, we select the weight matrix $A\in \mathbb{R}^{ns^2 \times ns^2}$ as the matrix form of a 2D convolutional operator. If we consider a circular convolution operator, then $A$ is a circulant matrix. Recall from Lemma 9 that $A$ can be parametrized as: $A = T - \mathrm{diag}(|T|\mathbb{1}) +\gamma I$. Conveniently $A$ is circulant iff $T$ is circulant. Therefore, the training problem for implicit CNNs can be cast as an unconstrained optimization problem using the above parametrization with a circulant $T$. We have implemented this framework of implicit CNN in our code and applied it to the CIFAR-10 dataset. In our preliminary numerical results, after 40 epochs with a batch size of 256, our model achieves 72.9% accuracy on unperturbed images while MON achieves 72.5% accuracy. We expect that the robustness improvements for our implicit neural networks compared to MON on the MNIST dataset will carry over to the implicit CNNs on the CIFAR-10 dataset. In the final version of the manuscript, we will add suitable plots comparing the accuracy and robustness of our implicit CNNs with MON in [2] on the CIFAR-10 dataset under the attack models mentioned in Appendix D.

**Runtime and number of iterations**: We believe that comparing the experimental results for the runtime of the training algorithm is of significance. It should be noted that in this paper, we propose an averaged iteration to compute forward and backward passes, compared to MON which uses the Peaceman-Rachford algorithm and the implicit deep learning model in [1] which uses Picard iterations. After submission of the manuscript, we observed empirically that we can accelerate the convergence using the same Peaceman-Rachford splitting algorithm as in MON. Our numerical experiments demonstrate that under our proposed well-posedness condition the Peaceman-Rachford algorithm converges. The rigorous study of convergence of the Peaceman-Rachford algorithm under our proposed well-posedness condition is a subject of an ongoing research.

We compared our runtime and number of function evals to MON and implicit deep learning in [1] on both MNIST and CIFAR-10 while training using a Tesla P100-PCIE-16GB GPU. Note that on CIFAR-10, both our model and MON use an implicit CNN architecture, while implicit deep learning only supports a fully-connected structure. The numbers are as follows:

1) our model with the averaged iterations on MNIST: average number of forward iterations was 12, average number of backward iterations was 13, and average time per epoch was 9.8 s
2) our model with the averaged iterations on CIFAR-10: average number of forward iterations was 10, average number of backward iterations was 10, and average time per epoch was 75.0 s
3) our model with Peaceman-Rachford on MNIST: average number of forward iterations was 8, average number of backward iterations was 9, and average time per epoch was 8.7 s
4) our model with Peaceman-Rachford on CIFAR-10: average number of forward iterations was 5, average number of backward iterations was 5, and average time per epoch was 89.7 s
5) MON on MNIST: average number of forward iterations was 17, average number of backward iterations was 16, and average time per epoch was 9.5 s
6) MON on CIFAR-10: average number of forward iterations was 5, average number of backward iterations was 5, and average time per epoch was 101.8 s
7) implicit deep learning on MNIST: average number of forward iterations was 10, average number of backward iterations was 5, and average time per epoch was 5.8 s
8) implicit deep learning on CIFAR-10: average number of forward iterations was 21, average number of backward iterations was 9, and average time per epoch was 61.2 s

In summary, on the MNIST dataset, our implicit model compared with MON has: larger number of iterations, faster iterations, and overall comparable training/evaluation runtime. On the CIFAR-10 dataset, our model compared to MON has: larger number of iterations, much faster iterations, and overall 25% shorter training/evaluation runtime.
___
**References**

[1]:El-Ghaoui et al. 2019. [Implicit deep learning](https://arxiv.org/abs/1908.06315)

[2]:Winston and Kolter 2020. [Monotone operator equilibrium networks](https://arxiv.org/abs/2006.08591)

[3]:Revay et al. 2020. [Lipschitz bounded equilibrium networks](https://arxiv.org/abs/2010.01732)

[4]:Pabbaraju et al. 2020. [Estimating Lipschitz constants of monotone deep equilibrium models](https://openreview.net/forum?id=VcB4QkSfyO)

[5]:Adly et al. 2014. [On one-sided Lipschitz stability of set-valued contractions](https://www.tandfonline.com/doi/abs/10.1080/01630563.2014.895760)

[6]:Ryu and Boyd 2016, [A primer on monotone operator methods](https://web.stanford.edu/~boyd/papers/pdf/monotone_primer.pdf)

[7]:Bauschke and Combettes 2017. [Convex analysis and monotone operator theory on Hilbert spaces](https://link.springer.com/book/10.1007/978-1-4419-9467-7)

[8]:Lim 1985. [On fixed-point stability of set-valued contractive mappings with applications to generalized differential equations](https://www.sciencedirect.com/science/article/pii/0022247X85903063)

[9]:Lohmiller and Slotine 1998, [On contraction analysis for non-linear systems](https://www.sciencedirect.com/science/article/abs/pii/S0005109898000193)

[10]:Davydov et al. 2021. [Non-Euclidean contraction theory for robust nonlinear stability](https://arxiv.org/abs/2103.12263)

---

> ### Author Response · Authors · 2021-08-10
> **Possibility of sharing additional documents**
>
>
> Several reviewers have asked us about the comparison between our well-posedness condition and the well-posedness condition in MON in [2]. We have generated several figures that compares these conditions on certain classes of $2\times 2$ matrices. Additionally, several reviewers asked us about the parametrization in Lemma 9 and enforcing the $\mathrm{osL}$ constraint in the code. In the initial submission we did not implement this parametrization but have done so since.
>
> We would be glad to provide this picture and an updated version of our code if there is an approved manner to share blind documents from authors to reviewers of NeurIPS 2021.

---

> > ### Comment · Area_Chair_jcVp · 2021-08-12
> > **About "May I include a link in the author response?"**
> >
> > External links are discouraged, but you may include a link if it is required to respond to a question from a reviewer. However, please remember that author responses may not contain any identifying information. Violating the double-blind reviewing policy in the author response may be grounds for rejection.  Furthermore, if you are including links to any external material, it is your responsibility to guarantee anonymous browsing.  As with supplemental material, reviewers are not required to read linked material in full, but may do so at their discretion.

---

> > > ### Author Response · Authors · 2021-08-12
> > > **Sharing links to additional documents**
> > >
> > > Thank you for your response. We will not share the picture nor the code unless directly asked

---

### Author Response · Authors · 2021-08-19
**Robustness results for CIFAR-10**

Per the request of several reviewers, we have run additional experiments for our model on CIFAR-10. We highlight the key results here:
___
For the FGSM attack on the CIFAR-10 dataset, we observe that our un-regularized implicit model is more accurate than MON for all amplitudes of perturbation. For example, at $\ell_\infty$-perturbation equal to $0.1$, the accuracy of our model is $39$ % whereas the accuracy of MON at this attack amplitude is $35$ %. Our regularized model, with the regularization parameter $\lambda =10^{-4}$
has a clean performance accuracy of $66$ % which is lower than the clean accuracy of both MON and our un-regularized model. However, our regularized model demonstrates a consistent improvement in accuracy for sizeable $\ell_\infty$-perturbations. For example, at an $\ell_\infty$-perturbation equal to $0.15$, the accuracy of our regularized model is $29$ % whereas the accuracy of MON at this attack amplitude is $24$ %. For the PGDM attack, we get comparable results to the FGSM attack with our un-regularized model outperforming MON for all perturbation and, for sizeable perturbations, our regularized model outperforming both MON and the un-regularized model.

---

### Decision · Program_Chairs · 2021-09-27

**Decision:**

Accept (Poster)

**Comment:**

The paper originally got 2 "Marginally above the acceptance threshold"s and 2 "Marginally below the acceptance threshold", all with relatively high confidences. The major challenges include a number of comparisons to existing works being lacking both in the theory and experiments, relatively weak experiments, missing some important details, etc. The authors did heavy rebuttals, including presenting extra experiments, and they seemed to take effect. Reviewer xt3G raised his/her score from 5 to 7. Reviewer ivrZ also raised to 7. Considering that the extra experiments and clarifications can be easily incorporated in the revision, the AC deemed that the paper is acceptable, thus recommended acceptance.